# Factors Related to Family Caregivers’ Readiness for the Hospital Discharge of Advanced Cancer Patients

**DOI:** 10.3390/ijerph19138097

**Published:** 2022-07-01

**Authors:** Ru-Yu Huang, Ting-Ting Lee, Yi-Hsien Lin, Chieh-Yu Liu, Hsiu-Chun Wu, Shu-He Huang

**Affiliations:** 1Department of Nursing, Mackay Memorial Hospital Tamsui Branch, New Taipei City 25160, Taiwan; scherrer.f383@mmh.org.tw (R.-Y.H.); wu1219.8770@mmh.org.tw (H.-C.W.); 2Department of Nursing, College of Nursing, National Yang Ming Chiao Tung University, Taipei 11221, Taiwan; tingting@nycu.edu.tw; 3Division of Radiotherapy, Cheng Hsin General Hospital, Taipei 11220, Taiwan; ch9145@chgh.org.tw; 4School of Medicine, National Yang Ming Chiao Tung University, Taipei 11221, Taiwan; 5Department of Health Care Management, National Taipei University of Nursing and Health Sciences, Taipei 11219, Taiwan; chiehyu@ntunhs.edu.tw; 6Department of Nursing, National Taipei University of Nursing and Health Sciences, Taipei 11219, Taiwan

**Keywords:** caregivers, home care services, hospitalisation, patient discharge, palliative care

## Abstract

*Background:* Many family caregivers of advanced cancer patients worry about being unable to provide in-home care and delay the discharge. Little is known about the influencing factors of discharge readiness. *Methods:* This study aimed to investigate the influencing factors of family caregivers’ readiness, used a cross-sectional survey, and enrolled 123 sets of advanced cancer patients and family caregivers using convenience sampling from four oncology wards in a medical centre in northern Taiwan. A self-developed five-point Likert questionnaire, the “Discharge Care Assessment Scale”, surveyed the family caregivers’ difficulties with providing in-home care. *Results:* The study showed that the discharge readiness of family caregivers affects whether patients can be discharged home. Moreover, the influencing factors of family caregivers’ discharge readiness were the patient’s physical activity performance status and expressed discharge willingness; the presence of someone to assist family caregivers with in-home care; and the difficulties of in-home care. The best prediction model accuracy was78.0%, and the Nagelkerke R^2^ was 0.52. *Conclusion:* Discharge planning should start at the point of admission data collection, with the influencing factors of family caregivers’ discharge readiness. It is essential to help patients increase the likelihood of being discharged home.

## 1. Introduction

In Taiwan, National Health Insurance pays for the inpatient medical expenses of advanced cancer patients. Previous studies had found that more than 40% of advanced cancer patients were hospitalised for more than 14 days in the last month of life, and about 60% died in the hospital [1]. Rapid turnaround of hospitalized patients reduces the tightness of hospitalisation and medical care and accomplishes the patient’s wish to live at home, but results in limited time to prepare patients for discharge [2,3]. Readiness for hospital discharge has been identified as an outcome indicator of reasonable care in hospitals [4]. According to Galvin et al. [5], the four factors influencing readiness for general hospital discharge are physical stability, adequate care support, psychological ability, and adequate information and knowledge. Significantly, due to tumour metastasis or local invasion of vital organs, advanced cancer patients have multiple moderate to severe symptoms that affect their physical function and ability to perform daily activities and increase their need for care, which also affects their readiness for hospital discharge [6,7,8]. Furthermore, evidence suggests that family caregivers are essential to the patient’s hospital return [9,10]. For family caregivers of these patients with advanced cancer, the factors influencing readiness for patients’ hospital discharge are their patient’s physical functioning, their perceptions of self-competence in managing the patient’s symptom distress at home, the difficulties in responding to the patient’s care problems, and proper support for them to undertake the patient’s post-discharge care [5,11,12]. However, discharge preparations are not currently routinely in the hospital, and post-discharge support referrals are often poorly implemented [13,14,15]. 

Previous studies have highlighted that family caregivers can perceive a lack of adequate knowledge of and information on patient care and symptom management during the discharge process while preparing to care for advanced cancer patients with severe illness with poor performance status and an increased dependence on life care, which ultimately affects their readiness for the patient’s hospital discharge [12,16,17]. Furthermore, many family caregivers of advanced cancer patients experience post-discharge care burdens [18,19] and worry about being unable to provide symptom distress and emergency health event management at home [12,20,21,22,23,24,25]. Compared with hospitalisation, home care involves fewer human resources and professional skills and less equipment support. As a result, home caregivers face a care burden once the severity of the patient’s symptom distress increases or the home caregiver cannot afford the patient’s care needs at home [19,26,27,28,29,30,31,32,33]. If the family’s support system does not provide the family caregivers with sufficient care assistance, they will have poor readiness for their patient’s hospital discharge. Family caregivers may sometimes delay patient discharge from the hospital until the patient dies [34,35,36]. Even after the hospital discharge of an advanced cancer patient, family caregivers may transfer the patient from one ward to another, send the patient to another hospital after discharge, or send the patient back to the hospital soon after discharge [4,10,12,37] to avoid providing in-home care by themselves, resulting in long-term hospitalisation [38,39].

Under Taiwan’s National Health Insurance (NHI) programme, all citizens are covered for hospital treatments [40,41]. Under the situation in which hospital care is inexpensive and easy to apply. The family caregivers often delay or are unwilling to discharge for home care, tending to allow patients to remain in the hospital to receive hospitalised palliative care and extended hospitalisation [12,27], and the acute beds are occupied. Hence, other patients who need acute inpatient medical treatment would have no available beds. One study indicated that nearly 1/5 of the patients were rehospitalisation on discharge [5], resulting in a failure to accommodate advanced cancer patients’ wishes to spend more time or even die at home [20,42]. In particular, for patients with lower physical activity functioning and higher care dependency, their primary family caregivers are often both the care providers and decision-makers regarding hospital discharge [25]. Therefore, family caregivers’ readiness for discharge and perceived difficulty with in-home care could be significant determinants of family caregivers’ decision-making of hospital discharge for home care [43]. Therefore, this study aimed to investigate the significant factors influencing family caregivers’ readiness for advanced cancer patients’ hospital discharge as a reference for future clinical implementations of hospital discharge planning and the possibility of home care support.

## 2. Materials and Methods

### 2.1. Design and Participants

This descriptive correlational study implemented a cross-sectional survey with self-designed questionnaires and two patient physical activity performance scales as the research tools. We selected the patients and their family caregivers through convenience sampling with the following criteria: patients diagnosed with advanced cancer and assessed by a physician could be discharged into home care. The family caregivers should mainly be the patients’ relatives, who are responsible for the patient’s home care after discharge and are the significant decision-makers. The patients and their families who met the above requirements were enrolled to participate in the research from four oncology wards in a medical centre (a contracted medical institution of the National Health Insurance) with more than 1000 beds in northern Taiwan. Advanced cancer patients who resided in long-term care institutions before hospitalization or the patients and their families who refused to participate in the research were excluded from the study. The study calculated the sample size as 123 sets of participants with the G POWER 3.1.9.7 linear multiple regression fixed model [44] with power = 0.80, *α* = 0.05, medium effect size = 0.15, and 11 independent variables. The participants were recruited for the study from 1 January 2019 to 31 December 2019; thus, 123 sets of participants were ultimately enrolled and completed the questionnaires. Afterward, the study followed the patients’ discharge status via hospital discharge data one month after the participant interviews.

### 2.2. Ethical Considerations

The authors of this study respected and protected the rights and interests of the participants. Therefore, we applied for and received approval from the Institutional Review Board of Mackay Memorial Hospital (Approval no. 18 MMHIS167e) before the survey and followed the guidelines of the Declaration of Helsinki. Then, the first author explained the research purpose and process to the participants. Afterward, we obtained informed consent forms from participants before administering the questionnaire. The questionnaire was collected only after participants provided their informed consent and signed a consent form. Any records related to personal privacy will be kept strictly confidential.

### 2.3. Instruments

#### 2.3.1. Demographic Characteristics and Patients’ Physical Functioning Assessment

Demographic data, awareness of illness truth, and expressed willingness to discharge to their family caregiver were collected through patient demographic questionnaires. Family caregivers’ demographic data, kinship or relationship with the patients, employment status, whether care was assisted by others, the financial burden of patient care cost, and readiness for hospital discharge were collected on the family caregiver questionnaire.

Patients’ physical functioning was assessed by the first author and one primary nurse at the point of care with the Karnofsky score (KPS scale) [45,46,47] and the Eastern Cooperative Oncology Group Performance Status Scale (ECOG scale) [48,49]. The ECOG has six grade rates from 0 (normal functioning and performance) to 5 (dead). The KPS consists of 11 levels of performance and functioning rates from 0 percent (dead) to 100 percent (normal functioning). From the previous study, the KPS score can be interchanged with the ECOG [50] and more precise as a guide for palliative care discharge needs [50,51], and assess the patients’ self-care ability. KPS 0–40: unable to care for self and requiring institutional or hospital care, 50–70: able to live at home and needs a varying amount of personal care assistance, and 80–100: no special care needed. Advanced cancer patients with better physical function (higher KPS score) were more likely to be discharged home [52].

#### 2.3.2. Family Caregivers’ Difficulties in Providing In-Home Care

To survey family caregivers about their difficulties with home care, the study used a self-developed structured five-point Likert self-administered questionnaire, the “Discharge Care Assessment Scale”, which was modified from the “Caregiver Preparedness Scale (CPS)” [9,53] based on interview data from ten family caregivers collected prior to this study and the literature references to family caregivers’ difficulties with in-home care. The scores range from 1 (strongly disagree) to 5 (strongly agree), with higher scores indicating greater home care difficulties. The scale contains 30 items (physical dimension: five items; emotional dimension: 13 items; social dimension: 12 items) and was tested by five experts, resulting in a content validity index (CVI) of 0.96. The score range of the scale was 30–150 points. The Cronbach’s α for this study’s performance instrument’s reliability was 0.93, and the Cronbach’s α of the three domains were 0.87, 0.88, and 0.86, respectively.

### 2.4. Statistical Analysis

Logistic Regression is an easily interpretable classification statistic technique that gives the probability of occurring event determinants. Family caregivers were divided into two groups based on their responses to discharge readiness (ready group vs. unready group). An independent t-test and a chi-square test were used to analyse statistically significant differences in means/frequency between the two groups. A value of *p* < 0.05 was regarded as statistically significant [16]. Finally, a binary forward logistic regression analysis was used to identify the influencing factors of the family caregivers’ readiness for hospital discharge, and *p* < 0.05 was the inclusion criterion for the best prediction model. All analyses were done using SPSS 24.0 (SPSS Inc., Chicago, IL, USA).

## 3. Results

### 3.1. Patient and Family Caregiver Demographic Characteristics

The mean age of the patients was 60.78 years, the mean KPS score was 51.54, and the mean ECOG score was 2.46, indicating that most advanced cancer patients had limited physical function in their daily activities and poor self-care ability. The correlation between ECOG PS and KPS scoring was high (r = −0.927, *p* < 0.001). Most advanced cancer patients were aware of their illness condition, and 78.9% (97/123) had expressed their willingness to be discharged from the hospital at some point. Through the hospital discharge data of the patients’ actual discharge status at the time of hospitalisation, this study found out that 91 patients (74%) had been discharged home, 30 patients (24.4%) had died in the hospital, one patient was discharged prior, one patient was discharged prior to death, and one patient had been transferred to another hospital for continuing hospitalisation (Table 1). The mean age of the family caregivers was 49.11 years, and 65.0% (80/123) of them expressed their readiness for hospital discharge (Table 1).

### 3.2. Demographic Characteristics and Family Caregivers’ Readiness for Discharge

We applied chi-square statistics to examine the results. Family caregivers who were ready for hospital discharge (i.e., the ready group) associated with a higher proportion of those patients discharged home than the unready group. We also applied independent t-tests and chi-square statistics to examine differences in the means/proportions of the patients’ and family caregivers’ demographic data versus family caregivers’ readiness for hospital discharge among the ready and unready groups. Compared to the unready group, the ready group had a higher proportion of patients who expressed a willingness to be discharged from the hospital and better physical activity function, and the family caregivers in the ready group also had someone who could assist with in-home care (*p* < 0.05) (Table 1).

### 3.3. Difficulties with In-Home Care and Readiness to Discharge

The mean difficulty experienced by the family caregivers with in-home care was rated 3.20, reflecting moderate to severe problems. Further test analysis of the three domains of the family caregivers’ difficulty with in-home care showed that the “social domain” had the highest average score (3.42 ± 0.68). In contrast, the most difficult item was “I think that home care cannot provide emergency care at home for patients after hospital discharge” (4.26 ± 0.86). The family caregivers also indicated the highest-scored item in each of the other two domains; the highest-scored item in the physical domain was “I feel exhausted because of the patient’s unstable condition and repeated hospitalizations” (3.70 ± 1.22); and in the emotional domain, “I am afraid that the patient might have an emergency event at home that I cannot manage” (4.07 ± 1.05). The study results indicated that the ready group perceived lower difficulties with in-home care than the unready group (*p* < 0.001) (Table 2).

### 3.4. Factors Predicting Family Caregivers’ Readiness for Hospital Discharge

In the study, we used logistic regression analysis for sensitivity to the influencing factors of the family caregivers’ readiness for hospital discharge. We found that four factors significantly differed between the ready group and the unready group: whether the patient had ever expressed willingness to be discharged from the hospital, the patient’s physical activity functioning (KPS score or ECOG), the family caregiver having someone to assist with post-discharge in-home care, and the difficulties in providing in-home care (whole scale or three sub-scales). We used these four factors as predictors in the binary logistic regression analysis. Model 1 indicated that a one-point increase in family caregivers’ perceived difficulties with in-home care could reduce family caregivers’ readiness for hospital discharge by 6% (OR = 0.94), while a one-point increase in the patient’s KPS score could increase the family caregivers’ readiness for hospital discharge (OR = 1.03). Those patients who expressed willingness to be discharged from the hospital were 5.88 times more likely to be ready for hospital discharge than those who never expressed such willingness. Additionally, the ratio was 3.13 times higher for family caregivers who had someone to assist with post-discharge in-home care than for those who had no such help (Table 3). Model 1, with these four factors, provided the best prediction of family caregivers’ readiness for the hospital discharge of advanced cancer patients. The overall prediction accuracy was 77.7%, and the Nagelkerke R^2^ was 0.51; Model 2 replaced KPS with ECOG for analysis and had a similar result; the overall prediction accuracy was 76.4%, and the Nagelkerke R^2^ was 0.53; Model 3 used three sub-scales to substitute the whole scale of difficulties in providing in-home care, and the overall prediction accuracy was 78.9%, and the Nagelkerke R^2^ was 0.54. However, the physical and emotional dimensions of difficulties in providing in-home care were insignificant in the predicting model. We further kept only the social dimension in Model 4 for analysis, the prediction accuracy was 78.0%, and the Nagelkerke R2 was 0.52 (Table 3).

## 4. Discussion

Regression analysis identified the influencing factors of family caregivers’ readiness for post-discharge in-home care; the study discussion follows.

### 4.1. Influence of Patients’ Expression of Willingness to Be Discharged

This study found that approximately 80% of patients had ever expressed a willingness to be discharged from the hospital, and two-thirds of family caregivers also expressed readiness for discharge. This finding is consistent with previous study results showing that most advanced cancer patients prefer to be discharged to in-home care [20,30,54]. The patients’ expression of such willingness is the most crucial factor influencing family caregivers’ readiness for hospital discharge [43]. Most caregivers want to respect the terminal patient’s wishes [43,55]. However, nearly 20% of the patients had never expressed their willingness to be discharged home. This result indicates the importance of helping patients express their willingness to be discharged home, which may require the intervention of health professionals to help patients voice their wishes and preserve their autonomy [56,57].

### 4.2. Patients’ Physical Status Affects Family Caregivers’ Care Difficulty and Readiness for Discharge

In this study, family caregivers who demonstrated readiness for hospital discharge had patients with better physical activity performance status (KPS & ECOG) than those in the unready group. An increase in the severity of the patient’s symptoms not only influences the patient’s ability to perform daily activities but also increases family caregivers’ care burden [7,28,31,32,58,59] and their care needs [21,59]. When the patient’s physical functioning is declining, or the patient has become completely disabled due to advanced cancer progression, the family caregivers are more dependent on hospice or palliative care support [5,11,12]. Thus, these caregivers were less ready for their patients to be discharged from the hospital.

### 4.3. Family Caregivers’ Concerns Regarding in-Home Care

The study findings are similar to previous studies showing that family caregivers have difficulties providing post-discharge in-home care for patients with an incurable disease, unstable illness condition, or disability in daily living [22,23,27,43,57]. These difficulties increase family caregivers’ perceived care burden and difficulties in providing their patients with post-discharge in-home care [19,28,58,59,60]. Among the most challenging care burdens experienced by family caregivers were emergency events and care needs, which affected their readiness for their patient’s discharge home [22,23,27,31,32,61]. Such challenges are the reason previous studies have indicated that health care professionals should provide family caregivers with home care skills training [62,63], teach them how to manage patients’ emergencies or severe symptom distress [12,35,36], and provide home-based palliative care [63,64,65,66]. When there is insufficient support from health professionals in post-discharge home care, family caregivers often cannot cope with and manage their patient’s symptom distress and emergency events at home. As a result, family caregivers might send patients for rehospitalisation sooner than desirable [35]. Therefore, how health professionals provide timely assistance to family in-home caregivers may be a crucial factor, especially when caregivers are faced with advanced diseases and patients who are highly dependent on their care. Such assistance can increase family caregivers’ confidence and reduce care difficulty in providing post-discharge in-home care [24,65,66] and accommodate patients’ desire to live at home [30,43,65]. For the above reasons, medical professionals should provide assessment and management based on the needs of the patient and family caregivers, including thorough evaluation and guidance regarding the knowledge and skills related to post-discharge in-home care [35,43,63,67], and should establish a telecare system and referral/handover system providing instruction and consultation to support family caregivers in providing post-discharge in-home care [15,36,68,69,70]. Furthermore, transitional care programmes or home-based palliative care are practical implementations for advanced cancer patients and caregivers’ discharge planning [64,65,66]. In addition, this study found that family caregivers ready for discharge had a significantly higher probability of having someone to assist with post-discharge in-home care than the unready group. This study result is consistent with previous surveys; family caregivers with workforce support might have lower post-discharge care burdens [11,22,71]. In the clinical implementation, to help the patient and caregiving family members prepare for post-discharge care work and needs, palliative care professionals need to assess workforce support for post-discharge home care and refer family caregivers to social resources such as long-term welfare or long-term care services. For example, Long-Term Care 2.0 in Taiwan [71,72,73] can provide care services in the patient’s own home and encourage family caregivers to participate in family support groups, share their care burden experience, participate in social activities, and exchange care information [67,74,75]. Additionally, resources related to purchasing and renting medical assistive devices should be provided to help family caregivers care for patients at home more efficiently. These interventions may help family caregivers prepare for post-discharge home care and prevent rapid rehospitalisation or negative experiences of post-discharge home care, thereby influencing family caregivers’ readiness for the next stage of hospitalization discharge planning [8,21,58,76].

## 5. Implications

The findings in our study have several implications for Taiwanese healthcare policy and programs that may be relevant worldwide. For family caregivers, insufficient skills and knowledge regarding attending to disease symptoms and providing in-home patients’ emotional and care support may pose difficulties and challenges. Therefore, the medical professionals must address discharge planning for advanced cancer patients beginning at admission by conducting a structured and systematic assessment through questions, such as the Discharge Care Assessment Scale used in this study, that examine the opinion of the in-home care workforce, the patient’s daily activity status, and the family’s discharge readiness to collect and evaluate the factors that affect the patient’s discharge. In particular, it is important to assess patients’ willingness to be discharged from the hospital and encourage them to express their desire to their family caregivers. Moreover, home-based health care and medical policies, along with the number of and timing of home care team visits for advanced cancer patients, can be adjusted flexibly according to the patient’s individual needs and those of the family caregivers. With these in place, feasible and effective care for patients with advanced cancer could then be consistently provided.

## 6. Limitations

There are some limitations to the interpretation of the study results. First, this is a cross-sectional survey study, which can only infer the degree of correlation between variables but cannot infer causal effects. In addition, because these patients’ conditions were more severe, non-discharge may not reflect family caregivers’ readiness for patient discharge but the patient’s death in the hospital. Moreover, the study subjects were advanced cancer patients and their family caregivers in oncology wards in a medical centre in northern Taiwan. The study results may not be able to be extrapolated to areas without national public health insurance. Next, the study used a questionnaire that has not been validated. Finally, it is also possible that other variables not considered in this study could explain caregivers’ readiness for discharge.

## 7. Conclusions

In the study, we identified the factors influencing family caregivers’ readiness for advanced cancer patients’ hospital discharge: the patients’ physical activity performance status (KPS or ECOG), the patients’ willingness to be discharged, whether family caregivers had someone to assist in providing post-discharge in-home care, and difficulties with in-home care, especially the social dimension of difficulties regarding in-home care. The study results could provide references for the clinical implementation of discharge planning for advanced cancer patients. These include encouraging patients to express their willingness to be discharged home to their family caregivers, providing family caregivers with care skills training to improve their in-home care ability, and helping families search for workforce support, environmental facilities, and equipment for in-home care. Additionally, the health care system should provide timely medical or nursing care assistance for patients’ symptom management or emergency health event treatment to alleviate the difficulty experienced by family caregivers. In-home care may be helpful for advanced cancer patients transitioning smoothly to post-discharge home care, where the caregivers can receive adequate care support and the patients have a good quality of life at home. Furthermore, it prevents early rehospitalisation and decreases the medical burden.

## Figures and Tables

**Table 1 ijerph-19-08097-t001:** Demographic characteristics and family caregivers’ readiness for discharge.

Variables	Total (*n =* 123)	Caregivers’ Readiness for Discharge		t/χ^2^	*p* Value
		No (*n* = 43)	Yes (*n =* 80)		
*Patients*					
Age (yrs)	60.78 ± 11.85	59.81 ± 11.13	61.30 ± 12.25	−0.66	0.509
KPS	51.54 ± 23.58	39.30 ± 19.57	58.13 ± 23.01	−4.55	<0.001 *
ECOG	2.46 ± 1.26	3.09 ± 1.02	2.13 ± 1.26	4.62	<0.001 *
Education				5.29	0.071
≤Junior high school	66 (53.7%)	23 (34.8%)	43 (65.2%)		
Senior high school	31 (25.2%)	15 (48.4%)	16 (51.6%)		
College and above	26 (21.1%)	5 (19.2%)	21 (80.8%)		
Marital status				0.28	0.599
Married	81 (65.9%)	27 (33.3%)	54 (66.7%)		
Single	42 (34.1%)	16 (38.1%)	26 (61.9%)		
Awareness of illness truth					0.767
Incomplete understanding	13 (10.6%)	5 (38.5%)	8 (61.5%)		
Complete understanding	110 (89.4%)	38 (34.5%)	72 (65.5%)		
Has ever expressed willingness to be discharged home				21.07	<0.001 *
No	26 (21.1%)	19 (73.1%)	7 (26.9%)		
Yes	97 (78.9%)	24 (24.7%)	73 (75.3%)		
Discharged home				11.34	<0.001 *
No	32 (26.0%)	19 (59.4%)	13 (40.6%)		
Yes	91 (74.0%)	24 (26.4%)	67 (73.6%)		
*Family caregivers*					
Age (yrs)	49.11±12.33	51.79±9.94	47.63 ± 13.29	1.95	0.054
Relationship				5.38	0.068
Spouse	55 (44.7%)	22 (40.0)	33 (60.0)		
Son/Daughter	53 (43.1%)	13 (24.5)	40 (75.5)		
Others	15 (12.2%)	8 (53.3)	7 (46.7)		
Gender				2.99	0.084
Female	79 (64.2%)	32 (40.5)	47 (59.5)		
Male	44 (35.8%)	11 (25.0)	33 (75.0)		
Education of caregivers				1.67	0.433
≤Junior high school	25 (20.3%)	7 (28.0)	18 (72.0)		
Senior high school	48 (39.0%)	20 (41.7)	28 (58.3)		
College and above	50 (40.7%)	16 (32.0)	34 (68.0)		
Employment				0.15	0.698
No	60 (48.8%)	22 (36.7)	38 (63.3)		
Yes	63 (51.2%)	21 (33.3)	42 (66.7)		
Adjusted employment				1.36	0.244
No	66 (53.7%)	20 (30.3)	46 (69.7)		
Yes	57 (46.3%)	23 (40.4)	34 (59.6)		
Someone assists with post-discharge home care				4.10	0.043 *
No	40 (32.5%)	19 (47.5)	21 (52.5)		
Yes	83 (67.5%)	24 (28.9)	59 (71.1)		
Cost of patient care				1.73	0.188
Affordable	89 (72.4%)	28 (31.5)	61 (68.5)		
Not affordable	34 (27.6%)	15 (44.1)	19 (55.9)		

Note: Count and percentage expressed data except for age, KPS, and ECOG, defined by mean ± standard deviation. * *p*-value < 0.05 indicates a significant difference in the corresponding variable between the ready for discharge and unready for discharge groups; KPS = Karnofsky Performance Status Scale score, ECOG = Eastern Cooperative Oncology Group Performance Status Scale score.

**Table 2 ijerph-19-08097-t002:** Difficulties in providing in-home care and family caregivers’ readiness for discharge.

Difficulties in Providing Care at Home	Total (*n =* 123)		Caregivers’ Readiness for Discharge		*p* Value
			No (*n* = 43)	Yes (*n* = 80)	
	Rank	Mean ± *SD*	Mean ± *SD*	Mean ± *SD*	
*The average score of single items*		3.20 ± 0.62	3.60 ± 0.43	2.98 ± 0.59	<0.001 *
*Physical dimension*		3.13 ± 1.01	3.67 ± 0.83	2.84 ± 0.99	<0.001 *
I feel exhausted because of the patient’s unstable condition and repeated hospitalizations	1	3.70 ± 1.22	4.28 ± 0.88	3.39 ± 1.27	<0.001 *
Taking care of a patient at home makes me feel exhausted	2	3.25 ± 1.33	3.93 ± 1.10	2.89 ± 1.30	<0.001 *
I have sufficient physical strength to take care of the patient ^R^	3	3.16 ± 1.12	2.86 ± 1.08	3.33 ± 1.11	0.028 *
Because I am taking care of the patient at home, I do not sleep well	4	3.13 ± 1.35	3.81 ± 1.12	2.76 ± 1.32	<0.001 *
My physical condition does not allow me to take care of the patient	5	2.73 ± 1.26	3.21 ± 1.19	2.48 ± 1.22	0.002 *
*Emotional dimension*		3.37 ± 0.73	3.78 ± 0.49	3.15 ± 0.74	<0.001 *
I am afraid that the patient might have an emergency event at home that I cannot manage	1	4.07 ± 1.05	4.51 ± 0.67	3.84 ± 1.14	<0.001 *
When taking care of the patient at home, I feel anxious, thinking that I might face a critical event at any time	2	3.85 ± 1.12	4.30 ± 0.83	3.60 ± 1.19	<0.001 *
I am worried that the patient will be harmed because I am not familiar with the skills needed to care for patients	3	3.63 ± 1.10	4.02 ± 0.89	3.43 ± 1.16	0.002 *
I need to stay with the patient for a long time, which causes me to feel very stressed	4	3.37 ± 1.20	3.84 ± 0.84	3.13 ± 1.29	<0.001 *
I can bear the responsibility of caring for the patient at home ^R^	5	3.34 ± 1.11	2.95 ± 1.25	3.55 ± 0.98	0.004 *
I am nervous when operating medical equipment (such as a suction device, an oxygen machine, or a patient-controlled analgesia pump)	6	3.32 ± 1.17	3.86 ± 0.99	3.03 ± 1.16	<0.001 *
I am not familiar with catheter care, which causes me to feel very stressed	7	3.24 ± 1.10	3.72 ± 0.91	2.98 ± 1.11	<0.001 *
I am not familiar with wound care, which causes me to feel very stressed	8	3.24 ± 1.11	3.79 ± 0.99	2.95 ± 1.07	<0.001 *
Taking care of the patient reduces my rest time, which causes me to be less patient than before	9	3.07 ± 1.14	3.42 ± 1.05	2.89 ± 1.15	0.013 *
I do not understand the patient’s disease status, which causes me to feel terrified	10	2.96 ± 1.28	3.23 ± 1.27	2.81 ± 1.27	0.083
In the long-term care of the patient, I feel lonely, and I have no one to complain to	11	2.93 ± 1.16	3.21 ± 1.10	2.79 ± 1.17	0.054
I do not feel pressure to take care of the patient at home alone ^R^	12	2.33 ± 1.21	1.79 ± 0.91	2.61 ± 1.26	<0.001 *
Long-term care for the patient at home does not influence my mood ^R^	13	2.24 ± 1.07	1.98 ± 0.83	2.38 ± 1.16	0.030 *
*Social dimension*		3.42 ± 0.68	3.90 ± 0.50	3.18 ± 0.64	<0.001 *
I think that home care cannot provide emergency care at home for patients after hospital discharge	1	4.26 ± 0.86	4.63 ± 0.54	4.06 ± 0.93	<0.001 *
I do not know about care subsidies and do not have sufficient funding to care for the patient	2	3.49 ± 0.91	3.81 ± 0.76	3.31 ± 0.94	0.002 *
I think that space at home is insufficient to allow the patient to receive better care at home	3	3.25 ± 1.16	3.88 ± 1.14	2.91 ± 1.03	<0.001 *
I think that when caring for a patient at home, I am more able to take care of both the patient and other family members ^R^	4	3.11 ± 1.11	2.47 ± 1.03	3.45 ± 0.99	<.001 *
I think that I can get more rest when providing patient care at home compared to care in a hospital ^R^	5	2.91 ± 1.08	2.28 ± 0.98	3.25 ± 0.97	<0.001 *
I think that home is the best place to provide a patient with complete care ^R^	6	2.83 ± 1.08	2.07 ± 0.80	3.24 ± 1.00	<0.001 *
I am aware of home care information, which allows me to rest assured about the patient’s return home after discharge ^R^	7	2.74 ± 1.03	2.35 ± 0.87	2.95 ± 1.05	0.002 *
I think that taking care of the patient at home does not influence my social activities or work ^R^	8	2.70 ± 1.18	2.28 ± 1.08	2.93 ± 1.18	0.003 *
I think the workforce at home is sufficient to help me take care of the patient ^R^	9	2.46 ± 1.23	1.81 ± 0.76	2.81 ± 1.29	<0.001 *
I have sufficient social assistance resources to help me take good care of the patient at home ^R^	10	2.41 ± 1.02	2.05 ± 0.92	2.61 ± 1.02	0.003 *
I never worry about the cost of caring for the patient ^R^	11	2.36 ± 1.16	2.19 ± 1.18	2.45 ± 1.15	0.230
I have the necessary assistive devices (such as beds, wheelchairs, and oxygen machines) in my home environment, which facilitates my home care of the patient ^R^	12	2.36 ± 1.16	2.09 ± 1.04	2.50 ± 1.20	0.053

Note: ^R^ negatively worded items; * *p*-value < 0.05 indicates a significant difference in the corresponding variable between the ready for discharge and unready for discharge groups.

**Table 3 ijerph-19-08097-t003:** Factors predicting family caregivers’ readiness for discharge.

Model	Factors	*B*	*S.E.*	*OR*	95% *CI*	*p* Value
1	The patient has expressed willingness to be discharged home (yes vs. no)	1.77	0.64	5.86	1.69–20.32	0.005 *
	Patient’s performance status (KPS)	0.03	0.01	1.03	1.01–1.06	0.015 *
	Someone assists with postdischarge in-home care (yes vs. no)	1.15	0.57	3.17	1.03–9.73	0.044 *
	Difficulties in providing in-home care	−0.06	0.02	0.94	0.90–0.97	0.001 *
2	The patient has expressed willingness to be discharged home (yes vs. no)	1.84	0.63	6.26	1.82–21.61	0.004 *
	Patient’s physical functioning (ECOG)	−0.61	0.25	0.54	0.34–0.88	0.012 *
	Someone assists with postdischarge in-home care (yes vs. no)	1.23	0.59	3.41	1.07–10.83	0.038 *
	A total score of difficulties in providing care at home	−0.07	0.02	0.94	0.90–0.97	0.001 *
3	The patient has expressed willingness to be discharged home (yes vs. no)	1.74	0.65	5.71	1.60–20.31	0.007 *
	Patient’s physical functioning (ECOG)	−0.62	0.25	0.54	0.33–0.88	0.013 *
	Someone assists with postdischarge in-home care (yes vs. no)	1.11	0.60	3.04	0.94–9.77	0.062
	The physical dimension of difficulties in providing in-home care	−0.06	0.08	0.94	0.81–1.09	0.428
	The emotional dimension of difficulties in providing in-home care	−0.03	0.05	0.97	0.89–1.06	0.475
	The social dimension of difficulties in providing in-home care	−0.12	0.05	0.89	0.81–0.97	0.008 *
4	The patient has expressed willingness to be discharged home (yes vs. no)	1.83	0.65	6.26	1.76–22.23	0.005 *
	Patient’s physical functioning (ECOG)	−0.70	0.24	0.50	0.31–0.80	0.004 *
	Someone assists with postdischarge in-home care (yes vs. no)	1.17	0.58	3.21	1.02–10.08	0.046 *
	The social dimension of difficulties in providing in-home care	−0.14	0.04	0.87	0.80–0.95	0.001 *

Note: * Indicates *p*-value < 0.05; the Nagelkerke R^2^ of the four models were 0.52, 0.53, 0.54, and 0.52 respectively; B = unstandardized coefficient, CI = confidence interval, SE = standard error, OR = odds ratio.

## Data Availability

Data are not publicly available to preserve the confidentiality of participants; however, they are available from the corresponding author upon reasonable request.

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
