# Peer review of "Factors Related to Family Caregivers’ Readiness for the Hospital Discharge of Advanced Cancer Patients"

_ijerph, 2022, doi:10.3390/ijerph19138097_

Round 1

Reviewer 1 Report

The authors aimed to investigate the influencing factors of family caregivers’ readiness for advanced cancer patients’ hospital discharge using a correlational and cross-sectional study. They highlighted main factors affecting caregiver discharge readiness and clinical implications. This study provides interesting clinical guidelines to better support the discharge readiness of patients with advanced cancer. However, some methodological points should be clarified. I hope my comments will be received in the same way in which they are intended – as positive and constructive encouragement.

Introduction

The introduction is clear and well researched. I have two comments:

- I would recommend splitting the first paragraph into 2, perhaps starting from "Previous studies have highlighted that family caregivers can perceive" p.2. This 2nd part deals more with the subjective experience of caregivers.

- Regarding the objectives p.2, I think it may be important to specify what the authors mean by "influencing factors" (with explanations/literature and hypotheses): What type(s) of variables (socio-demographic, clinical variables, perceived difficulty of in-home care...)? Why target specifically "perceived difficulty of in-home care"? The authors should further emphasize the value of their study in relation to the literature.

Materials and Methods / results

1)      What do the authors mean by "expressed willingness to discharge"? Is it a choice expressed to the hospital team or to the family? Or only when filling out the questionnaire?

2)      The “Patients’ physical functioning” was an important variable in the study and could be further detailed. Would it be possible, for example, to add the range of the scores?

3)      As I understand it, the authors use a scale that they modified for the present study to assess the caregivers’ difficulties with home care. I think it is important to further justify this choice and to further clarify the procedure used. In addition, would it be possible to obtain more precision on the sub-scores of this scale? A definition of the sub-scores with examples of items would be welcome in this section.

4)      “Family caregivers were divided into two groups based on their readiness for hospital discharge (ready group vs. unready group)”: I suppose that this variable was assessed in a binary way (yes or no). Would it be possible to make this clear in the method, as this variable is particularly central to the study? Furthermore, I wonder why the authors did not choose to assess this variable with a Likert-type scale to obtain a continuous score that could have been more informative and precise.

5)      At this point in the article (“Statistical analysis”), the rationale for the statistical analyses is unclear. On first reading, I wondered if the authors only wanted to describe and compare the groups or if they considered sociodemographic variables as factors influencing caregivers’ readiness for discharge (objective). The authors did not clearly explain why they performed multiple stages of statistical analysis and why they chose to use independent t and chi-square tests, then logistic regressions. They should clarify this point and perhaps clarify their objectives consequently. The parallel between the objectives, the presentation of the statistical analyses and the presentation of the results should be very clear, especially for busy readers who need to find certain information quickly.

6)      The previous point also leads me to consider the number of statistical tests involving a risk of type 1 error and the adjustment of the significance level that might be required in the case of multiple comparisons (depending on the authors' main objective).

7)      It is not clear to me how the authors arrived at a model that takes 4 factors into account (p.6) (without ECOG, Discharged home…). In addition, why did the authors detail all of the “Difficulties in providing in-home care” items (Table 2) and then ignore them in the final model?

Discussion / Limitations

The authors could add in the limitations of the study the use of a questionnaire that has not been validated. It is also possible that other variables, not considered in the present study, could explain the caregivers’ readiness for discharge (see introduction: “For family caregivers of patients, the influencing factors of readiness for patients’ hospital discharge are their patient’s symptom stability, their self-competence in managing the patient’s symptom distress at home, adequate information and knowledge…”).

Author Response

Reviewer comments:

The authors aimed to investigate the influencing factors of family caregivers’ readiness for advanced cancer patients’ hospital discharge using a correlational and cross-sectional study. They highlighted the main factors affecting caregiver discharge readiness and clinical implications. This study provides interesting clinical guidelines to better support the discharge readiness of patients with advanced cancer. However, some methodological points should be clarified. I hope my comments will be received in the same way in which they are intended – as positive and constructive encouragement.

Authors’ response:

Thanks to the reviewers for carefully reviewing this article and providing professional advice.

Introduction

Reviewer Comments: The introduction is clear and well researched. I have two comments:

- I would recommend splitting the first paragraph into 2, perhaps starting from "Previous studies have highlighted that family caregivers can perceive" p.2. This 2nd part deals more with the subjective experience of caregivers.

Authors’ response:

Thanks for the suggestion; the article has been modified regarding your suggestion

Reviewer comments:

Regarding the objectives p.2, I think it may be important to specify what the authors mean by "influencing factors" (with explanations/literature and hypotheses): What type(s) of variables (socio-demographic, clinical variables, perceived difficulty of in-home care...)? Why target specifically "perceived difficulty of in-home care"? The authors should further emphasize the value of their study in relation to the literature.

Authors’ response:

Thanks for your comments and questions. We have revised the description of the “Influencing Factors” and the research subject “Perceived Difficulty of Home Care” in the text, see page 2 (As highlighted in yellow). It is that “For family caregivers of these patients with advanced cancer, the factors influencing discharge readiness of patients’ family caregivers were the patient’s physical functioning and symptom stability, their perceptions of self-competence in managing the patient’s symptom distress at home, adequate information and knowledge to respond to the patient’s care problems, and proper support for them to undertake the patient’s postdischarge care [5, 11-13]”. Please see the yellow highlighted in context.

Reviewer comments:

Materials and Methods/results

What do the authors mean by "expressed willingness to discharge"? Is it a choice expressed to the hospital team or to the family? Or only when filling out the questionnaire?

Authors’ response:

Thanks for the questions. The mean of patients’ "expressed willingness to discharge" was modified to '"expressed willingness to discharge to their family caregiver were collected through patient demographic questionnaires” to clarify the meaning. see page 3 (As highlighted in yellow)

Reviewer comments:

2)      The “Patients’ physical functioning” was an important variable in the study and could be further detailed. Would it be possible, for example, to add the range of the scores?

Authors’ response:

Thanks for the suggestions. The “Patients’ physical functioning” was an important variable in the study, and we have done further details. We added them to the text, which was” The KPS score range is wide (0-100 points) which can be interchange with the ECOG and more precise as a guide for palliative care discharge needs [51-52], and assess the patients’ self-care ability. KPS 0-40: unable to care for self and requiring institutional or hospital care, 50-70: able to live at home and needs a varying amount of personal care assistance, and 80-100: no special care needed. Advanced cancer patients with better physical function (higher KPS score) were more likely to be discharged home [53].” see page 3 (As highlighted in yellow)

Reviewer comments:

3)      As I understand it, the authors use a scale that they modified for the present study to assess the caregivers’ difficulties with home care. I think it is important to further justify this choice and to further clarify the procedure used. In addition, would it be possible to obtain more precision on the sub-scores of this scale? A definition of the sub-scores with examples of items would be welcome in this section.

Authors’ response:

Thanks for the suggestions. This research formulated this research question based on clinical phenomena; therefore, the existing research scales are challenging to meet the research aim. Thus, the researchers reviewed the developed and used scales as the basis and revised the scale items based on the clinic interview results of 10 caregivers. Five experts examined the content validity. This questionnaire also had strong internal consistency reliability in this study. We described the detailed process of modified and developed scale in 2.3.2. The subscales information of the questionnaire were also described in 2.3.2, such as "The scale contained 30 items (physical domain: five items; emotional domain: 13 items; social domain: 12 items) and was tested by five experts, resulting in a content validity index (CVI) of 0.96. The score range of the scale was 30-150 points. The Cronbach’s α for this study performance instrument’s reliability was 0.93, and the Cronbach’s α of the three domains were 0.87, 0.88, and 0.86, respectively." (AS yellow-highlighted in context)

Reviewer comments:

4)    “Family caregivers were divided into two groups based on their readiness for hospital discharge (ready group vs. unready group)”: I suppose that this variable was assessed in a binary way (yes or no). Would it be possible to make this clear in the method, as this variable is particularly central to the study? Furthermore, I wonder why the authors did not choose to assess this variable with a Likert-type scale to obtain a continuous score that could have been more informative and precise.

 Authors’ response:

Thanks for the questions and the suggestion. In the study, “Family caregivers were divided into two groups based on their readiness for hospital discharge (ready group vs. unready group)” due to the study design investigating the main influencing factors of discharge readiness. The participants we chose were the primary family caregivers of “patients diagnosed with advanced cancer and assessed by a physician could be discharged into home care” (see 2.1, yellow-highlighted). Therefore, determining readiness for discharge (yes or no) in this situation is an essential thing that this study wanted to explore, which is why we did not use a Likert-style scale to obtain continuous scores.

Reviewer comments:

5)      At this point in the article (“Statistical analysis”), the rationale for the statistical analyses is unclear. On first reading, I wondered if the authors only wanted to describe and compare the groups or if they considered sociodemographic variables as factors influencing caregivers’ readiness for discharge (objective). The authors did not clearly explain why they performed multiple stages of statistical analysis and why they chose to use independent t and chi-square tests, then logistic regressions. They should clarify this point and perhaps clarify their objectives consequently. The parallel between the objectives, the presentation of the statistical analyses and the presentation of the results should be very clear, especially for busy readers who need to find certain information quickly.

Authors’ response:

Thanks for the questions and the suggestion.

We revised to contents' description to “Logistic Regression is an easily interpretable classification statistic technique that gives the probability of occurring event determinants. Family caregivers were divided into two groups based on their responses to discharge readiness (ready group vs. unready group). An independent t-test and a chi-square test were used to analyse statistically significant differences in means/frequency between the two groups. A value of p < 0.05 was regarded as statistically significant [17]. Finally, a binary forward logistic regression analysis was used to identify the influencing factors of the family caregivers’ readiness for hospital discharge, and p<.05 was the inclusion criterion for the best prediction model. All analyses were done using SPSS 21.0 (SPSS Inc., Chicago, IL, USA).” Please see part 2.4. Statistical analysis (as highlighted in yellow).

Reviewer comments:

6)      The previous point also leads me to consider the number of statistical tests involving a risk of type 1 error and the adjustment of the significance level that might be required in the case of multiple comparisons (depending on the authors' main objective).

Authors’ response:

Thanks for the questions and the comments.

We calculated sample size depending on the critical quantities: the type I and type II error rates α and β, and usually α and β are fixed at 5% and 20%, respectively (as highlighted in yellow). After discussion, the researchers did not consider adjusting the significance level for Type 1 error.

Reviewer comments:

7)      It is not clear to me how the authors arrived at a model that takes 4 factors into account (p.6) (without ECOG, Discharged home…). In addition, why did the authors detail all of the “Difficulties in providing in-home care” items (Table 2) and then ignore them in the final model?

Authors’ response:

Thanks for the comments and questions. In the independent t-test, the statistical result showed that the patients’ physical functioning scores of KPS and ECOG were statistically significant between the two groups. Following the previous study suggestion, we choose KPS as the patients’ physical functioning dependent variable logistic regression analysis. Please see the description in the text 2,3,1 and the reference 53. Please see the yellow highlighted in context.

Reviewer comments:

Discussion / Limitations

The authors could add in the limitations of the study the use of a questionnaire that has not been validated. It is also possible that other variables, not considered in the present study, could explain the caregivers’ readiness for discharge (see introduction: “For family caregivers of patients, the influencing factors of readiness for patients’ hospital discharge are their patient’s symptom stability, their self-competence in managing the patient’s symptom distress at home, adequate information and knowledge…”).

Authors’ response:

Thanks for the suggestion; the article has revised the context and added, “Next, the study used a questionnaire that has not been validated. Finally, it is also possible that other variables not considered in this study could explain caregivers' readiness for discharge.” in the Limitation regarding your suggestion. Please see page 9 (as highlighted in yellow).

Reviewer 2 Report

Thank you for the opportunity to review the paper. Well done for completing and reporting this study. 

It focuses on an important area that highlights where hospital clinicians could pay attention to in the care of cancer patients close to the end of life. 

Please see additional comments and suggested edits (embedded) in the document attached. 

Round 2

Reviewer 1 Report

I thank the authors for taking into account the comments, their explanations and changes accordingly. However, in my humble opinion, some points are still lacking in clarifications.

Introduction

Reading the objectives (at the end of the introduction: “Therefore, this study aimed to investigate the factors influencing of family caregivers’ readiness for advanced cancer patients’ hospital dis-charge…”), I always wonder what this study really brings to the literature. The objective was to “investigate the factors influencing of family caregivers’ readiness for advanced cancer patients’ hospital discharge”. However, the authors cited several studies in the introduction that have already highlighted these factors: “For family caregivers of these patients with advanced cancer, the factors influencing discharge readiness of patients’ family caregivers were the patient’s physical functioning and symptom stability, their perceptions of self-competence in managing the patient’s symptom distress at home, adequate information and knowledge to respond to the patient’s care problems, and proper support for them to undertake the patient’s post-discharge care [5, 11-13]”. I think the authors should more clearly highlight the interest of their study (in a specific paragraph with their objectives). Did they want to confirm the literature? Were they addressing the same issue but with a different approach (if so, which one?)? Did they consider other factors than those in the literature (if so, which ones?)? These details should appear clearly at the end of the introduction, as should the categories of factors assessed in the study to facilitate the reading and understanding of readers who are not familiar with this field and the study.

In addition, I encourage authors to check their statements for consistency with the articles cited in the Introduction. For example, reference [5] (mentioned just above, “For family caregivers of these patients with advanced cancer, the factors influencing discharge readiness of patients’ family caregivers were … [5, 11-13]”): this study did not focus on “patients with advanced cancer” and “the factors influencing discharge readiness” but it focused on “older patients” and “support needs of family caregivers of older people after hospital discharge”.

Methods

I'm sorry but I still don't understand why the authors made “An independent t test and a chi-square test to analyze a statistically significant differences in means/frequency between the two groups” for Demographic characteristics (Table 1) and Difficulties in providing care at home (Table 2). Do these analyses test one objective (if so, which one?) or describe the two groups (ready and not ready)? This should be clearly stated.

The authors should also specify any variables (or groups of variables) that they considered for the binary forward logistic regression analysis. Are these the variables presented in Tables 2 and 3? For “Difficulties in providing care at home”, why did the authors use the overall score and not the sub-scores (or items as considered in the analyses in Table 2)? It is also important that the authors clearly name these groups of variables in their objectives and clearly justify their choices. In particular, the authors should clearly justify their choice between KPS and ECOG in the article, as they did in their response to my comment. At the same time, their argument makes me wonder why they evaluated the 2 when they knew that KPS could be preferred in the literature.

Author Response

Reviewer comments:

Introduction

Reading the objectives (at the end of the introduction: “Therefore, this study aimed to investigate the factors influencing of family caregivers’ readiness for advanced cancer patients’ hospital discharge…”), I always wonder what this study really brings to the literature. The objective was to “investigate the factors influencing of family caregivers’ readiness for advanced cancer patients’ hospital discharge”. However, the authors cited several studies in the introduction that have already highlighted these factors: “For family caregivers of these patients with advanced cancer, the factors influencing discharge readiness of patients’ family caregivers were the patient’s physical functioning and symptom stability, their perceptions of self-competence in managing the patient’s symptom distress at home, adequate information and knowledge to respond to the patient’s care problems, and proper support for them to undertake the patient’s post-discharge care [5, 11-13]”. I think the authors should more clearly highlight the interest of their study (in a specific paragraph with their objectives). Did they want to confirm the literature? Were they addressing the same issue but with a different approach (if so, which one?)? Did they consider other factors than those in the literature (if so, which ones?)? These details should appear clearly at the end of the introduction, as should the categories of factors assessed in the study to facilitate the reading and understanding of readers who are not familiar with this field and the study.

In addition, I encourage authors to check their statements for consistency with the articles cited in the Introduction. For example, reference [5] (mentioned just above, “For family caregivers of these patients with advanced cancer, the factors influencing discharge readiness of patients’ family caregivers were … [5, 11-13]”): this study did not focus on “patients with advanced cancer” and “the factors influencing discharge readiness”, but it focused on “older patients” and “support needs of family caregivers of older people after hospital discharge”.

Authors’ response:

According to the introduction literature [5, 11-13], it is pointed out that the possible influencing factors affecting the patient's discharge and the difficulty of discharge, including the patient's physical functioning and the various difficulties faced by the caregiver in the process of caring for the patient at home. The National Health Insurance backs Taiwan Medical, the cost of hospitalisation is low, and it is easy to apply hospitalised palliative care. Clinically, family caregivers of patients with advanced cancer often delay discharge or do not want to be discharged from the hospital. In this context, family members are the primary caregivers of patients and critical medical decision-makers. They are also essential personnel who influence the decision-making of patients' discharge and care at home. Therefore, it is necessary to explore the factors that affect family members' preparation for discharge, including patient factors, family factors, and family members' difficulty in caring at home, as a reference for clinical discharge preparation support, for evaluation before discharge care planning for advanced cancer patients and early diagnosis of possible risk factors. Families with difficulties preparing for discharge will be referred and assisted by care resources as soon as possible to help implement comprehensive home care for terminal cancer patients.

In addition, thanks to the reviewer for reminding us that the original reference [13] was about the discharge of the elderly. This study did not focus on the elderly patients and elderly family caregivers, so the literature has been deleted.

Reviewer comments:

Methods

I'm sorry but I still don't understand why the authors made “An independent t test and a chi-square test to analyze a statistically significant differences in means/frequency between the two groups” for Demographic characteristics (Table 1) and Difficulties in providing care at home (Table 2). Do these analyses test one objective (if so, which one?) or describe the two groups (ready and not ready)? This should be clearly stated.

Authors’ response:

Because we want to explore the factors that are significantly different between those who are ready to be discharged and those who are not, those factors are used as variables for further logistic regression analysis to explore the main influencing factors that affect family caregivers' readiness for discharge home. It can be used as a reference for future inpatients' discharge planning, reduce the difficulty of hospital discharge after re-hospitalization and help them take care at home.

Reviewer comments:

The authors should also specify any variables (or groups of variables) that they considered for the binary forward logistic regression analysis. Are these the variables presented in Tables 2 and 3? For “Difficulties in providing care at home”, why did the authors use the overall score and not the sub-scores (or items as considered in the analyses in Table 2)? It is also important that the authors clearly name these groups of variables in their objectives and clearly justify their choices. In particular, the authors should clearly justify their choice between KPS and ECOG in the article, as they did in their response to my comment. At the same time, their argument makes me wonder why they evaluated the 2 when they knew that KPS could be preferred in the literature.

Authors’ response:

Thanks to the reviewer for your advice and questions, which helped us think and revise further. Because the physical function evaluation of advanced cancer patients in Taiwan hospitals currently includes the patient's ECOG & KPS, to understand the physical function status of advanced cancer patients, two scales were included for evaluation, and the two scales were significantly correlated (r= -.927, p= <.001). There is indeed a significant difference between the two groups (ready & unready) in the physical function scores of the patients on these two scales. We are also grateful for the advice on conducting the three subscales for inclusion in the statistical analysis. Therefore, the two physical functional status scales of patients and the three dimensions subscales of the family caregivers’ difficulties in providing in-home care were included in the statistical analysis of the prediction module, and the results were modified and improved in Table 3. The study result showed that Model 4 only requires a 12-item social dimension scale, and the 6-Grade ECOG scale seems more accessible and streamlined and has higher applicability and efficiency for clinical works. Thank you so much!